# Habitual Diet Pattern Associations with Gut Microbiome Diversity and Composition: Results from a Chinese Adult Cohort

**DOI:** 10.3390/nu14132639

**Published:** 2022-06-25

**Authors:** Yuhan Zhang, Hongda Chen, Ming Lu, Jie Cai, Bin Lu, Chenyu Luo, Min Dai

**Affiliations:** 1Medical Research Center, Peking Union Medical College Hospital, Chinese Academy of Medical Sciences and Peking Union Medical College, Beijing 100730, China; zyhmsf426@student.pumc.edu.cn (Y.Z.); minglu@student.pumc.edu.cn (M.L.); lubin838744@student.pumc.edu.cn (B.L.); luochy23@163.com (C.L.); 2Department of General Surgery, Peking Union Medical College Hospital, Chinese Academy of Medical Sciences and Peking Union Medical College, Beijing 100730, China; caijie1113@126.com

**Keywords:** gut microbiota, habitual diet, 16S rRNA gene sequencing, Chinese, adults

## Abstract

The influence of long-term diet on gut microbiota is an active area of investigation. The present work aimed to explore the associations between habitual diet patterns and gut microbiota in a large sample of asymptomatic Chinese adults. The gut microbiome was profiled through the sequencing of the 16S rRNA gene in stool samples from 702 Chinese adults aged 50–75 years who underwent colonoscopies and were diagnosed to be free of colorectal neoplasm. Long-term dietary consumption was assessed through a food-frequency questionnaire. The microbial associations with specific food groups and the posteriori dietary pattern were tested using the Kruskal–Wallis H test, permutational ANOVAs, and multivariate analyses with linear models. The Shannon indexes generally shared similar levels across different food intake frequency groups. Whole grain and vegetable intakes totally explained 1.46% of the microbiota compositional variance. Using the data-driven posteriori approach, a general dietary pattern characterized by lower intakes of refined grains was highlighted to be associated with higher abundances of the genus *Anaerostipes* and a species of it. We also observed 17 associations between various food group intakes and specific genera and species. For instance, the relative abundances of the genus *Weissella* and an uncultured species of it were negatively associated with red meat intake. The results of this study support the idea that the usual dietary consumption measured by certain food items or summary indexes is associated with gut microbial features. These results deepen the understanding of complex relationships of diet and gut microbiota, as well as their implications for gut microbiome studies of human chronic diseases.

## 1. Introduction

The gut microbiota play a vital role in the host homeostasis maintenance, ranging from the catabolism and biosynthesis of essential nutrients to immune regulations and nerve signals transmission [1,2,3]. Pathological alterations of the gut microbiota community have been shown to be involved in the development of a wide spectrum of health disorders [4,5,6,7]. Various human lifestyle and physiological variables exert differential impacts on the gut microbiota throughout the life span, with environmental factors outweighing the genetic ones [2,8]. Among these environmental variables, including living behaviors, food habits, and medication, diet has been a primary research focus recently due to its diversity and easily modifiable properties. Food intake is increasingly considered as an intervention target for disease treatment and health promotion, and it has further evolved into a hot research area called precision nutrition [9,10,11]. Additionally, inter-individual heterogeneity in gut microbiota mainly arising from differences in personal physiological and lifestyle variables (such as diet) may confound microbiota analyses, resulting in spurious associations in gut microbiome studies of human diseases [12,13].

Short-term dietary changes such as the introduction of specific nutrients, foodstuffs, or special diet patterns can rapidly and significantly influence gut microbial profiles [14]. The observed transient effect supports diet’s causal role in gut microbiome alterations while necessitating the study of dietary habits’ impact on the gut microbiota in the long run because previous studies have either focused on single nutritional factor at a time or only substances without deleterious effects on humans [15]. Additionally, specific changes induced by short-term interventions generally do not persist due to their limited duration, whereas long-term dietary habits may dominantly drive gut microbiota composition [14,15]. Large-scale observational studies have accordingly investigated the associations between usual diet and gut microbiota composition, unveiling the relationships between food intakes and the gut microbiota profiles, as well as some particular dietary patterns. Studies have suggested that plant-rich food intakes are associated with a more diverse and compositionally distinct microbiota, as well as elevated abundances of specific bacterial taxa with a greater potential to produce short chain fatty acids (SCFAs), including fruits, fiber-rich breads, and vegetarian or Mediterranean diets [16,17]. By comparison, the Western diet and high intakes of animal protein have been reported to be associated with lower microbiome diversity and the enrichment of harmful bacteria [16,18].

However, previous studies mainly focused on either some particular food groups [19,20,21], such as fiber, red meat and processed meat, or on Western-population-oriented predefined diet quality scores, such as the Healthy Eating Index [13,22,23] and the Mediterranean Diet Score [13,23,24]. Habitual dietary variables are multidimensional, with internal correlations. Summary dietary indices can simplify complexity by quantifying dietary variance in a single measure and possibly offer a potential means of diet control in microbiota studies. In addition, caution should be taken in extrapolating findings from European and American populations to other ethnic groups. To our knowledge, only two published studies have specifically looked into this topic among Chinese populations. Yu et al. observed that the long-term diet quality was positively associated with fecal microbiome diversity and an abundance of fiber-fermenting bacteria among people lived in urban communities in a single region (Shanghai, China) [25]. Lu et al. provided a nationwide gut microbiota baseline of the Chinese population and knowledge on important environmental covariates, though with a sole focus on the dominant staple food type (including rice and wheat) [26]. There remains great uncertainty with respect to the long-term dietary habits related gut microbiome profiles fluctuations among Chinese people, especially those over 50 years old who are prone to chronic diseases with the potential participation of the gut microbiota.

The aim of the present study was to explore the dietary associations of the posteriori long-term diet pattern and habitual food intakes with gut microbiota composition in a large sample of asymptomatic individuals aged 50–75 years from six cities of China.

## 2. Methods

### 2.1. Study Participants

This study was based on the TARGET-C study initiated in May, 2018. The rationale, design, and protocol have been published and extensively described elsewhere [27,28,29]. Briefly, the primary objective of the TARGET-C study was to compare the effectiveness of the colonoscopy-based fecal immunochemical test (FIT) and risk-adapted triage screening strategies for colorectal cancer in China. Epidemiological data and biological samples collected during this study were also used for interested investigations, such as the work presented here. After obtaining signed informed consent, the eligible participants were randomly assigned into three groups to undergo colonoscopy, FIT, and risk-adapted colorectal cancer screening (i.e., the colorectal cancer risk assessment followed by FITs for the low-risk group or colonoscopies for the high-risk group). Patients who had positive FIT results were also required to undergo a subsequent colonoscopy. All participants undergoing colonoscopy were required to collect stool samples within 24 h prior to bowel preparation for colonoscopy. This study was approved by the Ethics Committee of the National Cancer Center/Cancer Hospital, Chinese Academy of Medical Sciences, and Peking Union Medical College (18-013/1615).

For the present study, we included participants who had no abnormal findings at screening colonoscopy and had available stool samples for microbiota sequencing. Exclusion criteria comprised a history of cancer and any current administration of anticoagulants, analgesics, and anti-rheumatic drugs. In addition, patients exhibiting abnormal abdominal symptoms, such as abdominal pain, diarrhea, constipation, and hematochezia, within 1 month before the colonoscopy examinations were excluded. More details of the participants’ enrollment can be found in Appendix A.

### 2.2. Stool Sample Collection

Eligible participants for colonoscopy were instructed to collect two stool samples at home prior to bowel preparation for the scheduled colonoscopy within 24 h. One was collected using the FIT stool collection device for extended microbiome analysis. Existing evidence suggests that feces collected by these devices are stable at room temperature and can be used for gut microbiota studies [30]. The stool-filled containers in storage boxes were delivered to a central laboratory and immediately frozen at −20 °C until DNA extraction. In this study, we used these stool samples for 16S rRNA sequencing. The other collected stool samples were kept in stool container tubes, then packaged in insulated boxes equipped with ice packs, and brought to the clinical sites on the days of the colonoscopies. On receipt, the fecal samples were frozen at −80 °C and subsequently transported by a cold chain to the central biobanks for further research.

### 2.3. DNA Extraction and 16S rRNA Gene Sequencing

DNA was extracted using the QIAamp Fast DNA Stool Mini Kit (QIAGEN). The V4 region of the microbial 16S rRNA gene was amplified and sequenced on the Illumina MiSeq sequencing platform. To avoid end-read sequencing errors, all reads were truncated at the 150th base and a median Q score of >20. Noisy sequences, chimeric sequences, and singletons were removed, and then amplicon sequence variants (ASVs) were inferred from the clean sequencing reads using the DADA2 pipeline built into Qiime2 [31]. Taxonomy was assigned to each ASV using the classify-sklearn classification methods via the q2-feature-classifier plugin built from the Greengenes database (release 13.8). To quantify the taxonomic composition, all sequences were rarefied to an even sampling depth of 10,000. Only the taxa and taxa present in at least 1% of the samples with an average relative abundance greater than 0.01% were included in the downstream analyses. Diversity metrics were calculated using the R package vegan, including α-diversity index and distance-based β-diversity. The relative abundances of each taxon were used in the following analyses.

### 2.4. Dietary Data Collection

Information about food intake during the past 12 months was collected through a food-frequency questionnaire (FFQ). Dietary data covered 9 major food groups in China: red meat (pork, beef, lamb, etc.), white meat (fish, chicken, duck, goose, etc.), eggs, dairy products, cooked meat (e.g., sausage), refined grains (rice, wheat, etc.), whole grains (millet, corn, sorghum, etc.), fresh fruits, and fresh vegetables. All these foods were examined with 5 frequency levels of habitual consumption (monthly or never/rarely, once a week, more than 1 time per week, daily, or more than 1 time per day) during the past 12 months. For analysis purposes, we transformed the frequency to times per week (i.e., 0, 1, 4, 7, and 14, respectively).

### 2.5. Dietary Pattern Analysis

Posteriori dietary patterns were derived from the 9 food groups using factor analysis with a principal component method. We applied a factor analysis with the principal component method to identify the major common factors. Orthogonal varimax rotation was performed to attain mutually independent structure with great interpretability. The optimal number of factors was determined by the scree plot examination of the true dataset compared to random “parallel” matrices, factor interpretability, and the variance explained (5%) by each factor. Finally, we chose the three-factor solution, totally explaining 50% of the whole variance of food intake frequencies (see Appendix A). Using the k-means clustering method, we finally clustered the participants into 3 groups according to the weighted factor scores from the factor analysis. For more details, see Appendix A.

### 2.6. Statistical Analysis

Covariates, including sociodemographic variables (sex and age), lifestyle factors (cigarette smoking, alcohol drinking, and physical activity) and BMI (in kg/m^2^) were adjusted in the diet–microbiome association analysis. Distributions of ASV-based alpha-diversity (including Shannon, richness, chao1, Simpson, Pielou, ACE, and faith_pd index) by different food intake frequency groups were compared using the Kruskal–Wallis H test. Associations between dietary variables and the β-diversity dissimilarities were evaluated using a permutational multivariate ANOVA (PERMANOVA, 999 permutations ) with adjustment for covariates, which was also used to measure the percentage of variation in microbial composition explained by the dietary variables. A *p*-value of <0.05 was considered to be significant. For a better visualization of the interindividual variation in gut microbiota composition, unconstrained principal coordinate analyses (PCoAs) of the Bray–Curtis distance were plotted and color-coded based on sex, age group, and BMI. Associations between dietary variables and gut microbiome profiles at the relative abundances of phyla, genera, and species level were tested using multivariate associations with linear models (MaAsLins). Detailed information regarding MaAsLins is provided in Appendix A. Models were multi-adjusted for the aforementioned covariates with a BH-adjusted *p*-value of <0.1 considered significant. All analyses were performed using R Version 4.0.5.

Although gut microbiota have been widely reported to geographically vary [32], it is hard to dissect the mixed effects of, for instance, lifestyle and long-term diets captured by the geographical variable. Thus, we conducted a sensitivity analysis among participants from the same province instead of regarding geography as a covariate to be adjusted, and we also considered the sample size. Additional sensitivity analyses were conducted by excluding (1) 223 participants who were assessed as at high risk of colorectal cancer or had positive FIT results and (2) 360 participants with BMI < 18.5 kg/m^2^ or >24.0 kg/m^2^.

## 3. Results

### 3.1. Study Sample Characteristics

A total of 702 participants were included in our final analysis, including 369 women and 333 men. Characteristics of the study population are presented in Table 1. The majority of the included individuals were aged between 50 and 70 years old, and they were evenly distributed by an age interval of 5 years, with only 5.56% aged over 70. The proportion of current smokers was 73.36%. Nearly two thirds of the population were non-drinkers. The BMI values were regrouped into three groups according to the Chinese definitions of “overweight” and “obesity”, with more than a half having a BMI of less than 24 kg/m^2^.

The usual dietary consumption of the participants is presented in Table 2. The amount of physical activity was evaluated using metabolic equivalent hours per day (MET-hours/day), which was regrouped into quantiles. The geographical distribution is also presented. The usual food intakes frequencies of the participants are presented in Table 2. Microbiota composition showed great interindividual variability at the phylum level (see Figure 1).

### 3.2. Data-Driven Posteriori Dietary Patterns

Three dietary patterns were identified in the present Chinese population (Appendix A). The first cluster, a traditional dietary pattern of the Yangtze River Delta, represented a typical traditional diet in South China characterized by high intakes of refined grains and vegetables but low intakes of cooked meat. A majority of participants from two sites of Zhejiang province, part of the Yangtze River Delta, followed this traditional Yangtze River Delta dietary pattern (indicated as Cluster A; see Appendix A). The second cluster was a modern dietary pattern that was characterized by specifically high intakes of eggs, dairy, fruits, vegetables and whole grains accompanied by medium intakes of red meat and white meat (indicated as Cluster B). The third cluster, labeled as the general dietary pattern, was characterized by the generally higher intake of each food group (4–6 times per week), except for the relatively lower consumption of cooked meat, compared to the other dietary patterns (indicated as Cluster C).

### 3.3. α-Diversity Indexes Distributed by Food Intake Frequencies

For the Shannon index, no significant differences were observed among different food intake frequencies for the nine food groups (Figure 2). Regarding red meat, white meat, cooked meat, dairy products, whole grains, and vegetables, the α-diversity index shared similar levels across different food intake frequency groups (Appendix A). The richness, chao1, ACE, faith_pd index were significantly distributed by egg intake frequencies (Appendix A). For refined grain and fruit consumption, the faith_pd index presented different distributions among different food intake frequency groups (Appendix A).

### 3.4. Associations between Dietary Variables and β-Diversity

Unconstrained PCoAs of the Bray–Curtis distance are shown in Figure 3. Compositional dissimilarities (β-diversity) of the gut microbiota between men and women and across different BMI groups were detected (Figure 3A,C). Although no clear clustering appeared among age groups, a grouping pattern along the gradient of age groups could be observed (Figure 3B). Arrows indicate the direction of gradient for covariates and were obtained via the envfit function (package “vegan”). Figure 3D presents the associations between dietary variables and β-diversity matrices found using PERMANOVAs. The Bray–Curtis distances of inter-individual dissimilarities were associated with whole grains and vegetables, explaining 1.46% of the total variation in the gut microbiota composition measured by the partial R^2^ value with age, sex, BMI, smoking, alcohol consumption, and physical activity adjusted.

### 3.5. Associations between Dietary Variables and Relative Abundances of Taxa

Taxa significantly associated with food groups and the posteriori dietary pattern are presented in Table 3. For instance, the genus *Weissella* and an unknown species of it were negatively associated with weekly red meat intake. Cooked meat was positively associated with an abundance of the genus *Coprobacter*. The relationships of *Weissella* and *Coprobacter* were kept consistent in the sensitivity analyses by restricting participants from a single province or removing individuals at a high risk of intestinal diseases, respectively (Appendix A).

Dairy intake was positively associated with the genus *Anaerostipes* and an unknown species of it. Moreover, we found significant positive associations for whole grain intake with a species of the genus *Megasphaera* and refined grain intake with a species of the genus *Lactobacillus*, which were also observed in the sensitivity analyses (Appendix A). Vegetables were negatively inversely associated with the genus *Eubacterium coprostanoligenes* group and a species of it, a species of the genus *Christensenellaceae R7* group belonging to the family *Christensenellaceae*, and the genus *Leuconostoc*. For the whole picture of the habitual food intakes, individuals leading the general dietary style (Cluster C) had higher abundances of the genus *Anaerostipes* and a species of it compared to those who had the traditional Yangtze River Delta dietary pattern (Cluster A) characterized by higher intakes of refined grains and vegetables and lower intakes of dairy products.

## 4. Discussion

In this population-based study of 702 healthy Chinese adults free of colorectal neoplasm aged 50–75 years, we examined the associations between the habitual dietary pattern and the gut microbiome. Our data revealed that the α-diversity index generally shared similar levels across different food intake frequencies among nine major food groups, whereas whole grain and vegetable intakes drove the dissimilarities in gut microbial composition, as indicated by the distance-based β-diversity dissimilarities. Based on the data-driven posteriori dietary pattern analyses, our results also highlighted the relationship of the general dietary style with higher abundances of the genus *Anaerostipes* and a species of it, which was characterized by lower intakes of refined grains. Moreover, we observed a number of positive or inverse associations between usual food groups and abundances of certain taxa, concentrated in genera within the phylum *Firmicutes*.

Previously reported evidence supports our findings. Evidence from a randomized diet intervention trial aiming to examine the effect of carbohydrate type on gut microbial composition and function and metabolites showed that *Anaerostipes* had a higher abundance after a simple carbohydrate diet compared to a refined carbohydrate diet [33]. Due to the role of *Anaerostipes* as a butyrate producer, low abundance after the consumption of refined carbohydrate foods may contribute to the unfavorable effects of diets rich in refined carbohydrates. In addition, the authors of a recent study reported a myo-inositol pathway in *Anaerostipes spp.*, which was most abundantly present in mammalian tissues and fruits, suggesting a newly discovered benefit of intestinal *Anaerostipes spp.* for host health promotion [34]. In our study, participants consuming general diets had higher weekly fruit intakes than individuals with the traditional Yangtze River Delta dietary pattern.

For the specific food groups, our results showed that genus *Weissella* and an unknown species of it were negatively associated with weekly red meat intake. *Weissella* is a member of the lactic acid bacteria group, which has been well-studied and is best known for its potential in imparting beneficial human health effects [34]. Some strains of *Weissella* can prevent lipopolysaccharide-induced proinflammatory stress in murine macrophages and human colonic epithelial cells [35]. Dairy has presented a positive association with the abundance of *Anaerostipes,* which warrants further investigation, whereas mice model studies have suggested that *Anaerostipes caccae* may be involved in the protective process against the allergic response to cow’s milk [36]. The association between vegetable intake and *Christensenellaceae* disappeared after excluding individuals with abnormal BMI levels, predominantly overweight and obese people. This phenomenon could be explained by previously reported evidence that suggests that the relative abundance of *Christensenellaceae* in the human gut is inversely related to host BMI in different populations, making its relationship with BMI the most robust and reproducible link between the microbial ecology of the human gut and metabolic disease [37].

The *Eubacterium coprostanoligenes* group is characterized as one of the hub genera in the fecal micro-ecosystem of high-fat diets, and studies have shown that the *Eubacterium coprostanoligenes* mediates the effect of high-fat diets on dyslipidemia through sphingosine [38]. The requirement of lecithin for the growth of *Eubacterium coprostanoligenes* [39], which is primarily rich in animal foods, may partly explain the negative association between the relative abundances of *Eubacterium coprostanoligenes* and vegetable intakes found here.

In the present study, we used aggregated items to collect information on broad dietary habits of participants for the sake of convenient dietary data collection. This led to high variability in terms of specific food types and nutrient composition, as well as the population-specific findings. For example, people residing in Europe consume different types of vegetables than Chinese people. Moreover, the complexity of food composition including macronutrients, micronutrients, and food additives made it difficult to elucidate the intricate diet–microbiota relationship. The significant findings in our study need to be cautiously interpreted, and some associations could be explained from a biological mechanistic standpoint. Thus, additional efforts and deeper insights regarding the underlying mechanisms are required before considering translating such knowledge to personalized diet intervention strategies. Nevertheless, we have confirmed that future studies should consider dietary variables as covariates in analyses of disease-microbiome associations to disentangle the effects of diet on the gut microbiome from disease-related associations. To simplify the complexity of multidimensional diet data with internal correlations, researchers can the dietary index as a summary measure when quantifying dietary variance in microbiota studies instead of individual dietary features [13], including priori or posteriori dietary indices [40].

The presented study is the so-far largest multi-center study of the association between the gut microbiota and the habitual diet with unitary and general measurements in the Chinese population. However, some points should be considered in interpreting our findings. Firstly, we only assessed nine commonly consumed food groups in the Chinese diet using aggregated items, so food groups that could be further classified were broadly considered, e.g., milk and yogurt were considered as general dairy products. Although the major dietary patterns in the studied Chinese adults were well-captured, only the frequencies (not the quantities of the major food groups) were collected, which made it less feasible to completely quantify food intakes. In addition, given the potentially rapid and transient effects of food on the human gut microbiome [14], the bacterial profiles characterized in a single fecal sample will likely reflect the effect of food consumption patterns in the period immediately prior to sample collection and not necessarily a participant’s long-term steady state. Additionally, long-term dietary habits were coarsely assessed using a one-time FFQ that collected data on food intake patterns over the prior 12 months. Thus, large-scale observational studies using accurate frameworks to capture long-term dietary exposures and stable gut microbiota composition and to reduce random within-person variation are needed for exploration of associations between habitual food intakes and gut microbiome. Subsequent time points with both dietary and microbiota data would be of utmost interest to investigate the stability of the studied relations over time. Though the participants of our study were from multiple regions, we performed a sensitivity analysis with individuals from a single province instead of regarding region as a covariate since the dietary information partly captured the geographical characteristics of the studied population (Appendix A). However, we cannot rule out residual confounding effects due to imperfectly measured covariates and unmeasured confounders, despite multivariable adjustments and sensitivities analyses. Furthermore, the TARGET-C study was initially established to evaluate the effectiveness of different colorectal cancer screening strategies. Participants enrolled in this study were apparently healthy upon recruitment according to stringent inclusion criteria but no systematic physical examination, thus providing a less pure foundation to investigate the diet–microbiota relationship. Extensive studies in a completely disease-free context are needed. Finally, the annotation resolution of the 16S rRNA amplicon sequencing was limited, so future efforts focusing on a broader picture of microbiome variability and the potential functional capability of the gut microbiome through shotgun metagenomics may provide deeper insight into the diet–gut microbiome relationship.

## 5. Conclusions

In summary, in a large sample of the Chinese population free of colorectal neoplasm, we found that the long-term dietary pattern characterized by lower intakes of refined grains was associated with higher abundances of the genus *Anaerostipes* and a species of it. The dietary pattern can act as a summary measure that captures gut microbiota variance attributable to habitual diet in microbiome studies. Future studies are needed to investigate whether and to what extent the gut microbiota may mediate or modify the effects of habitual diets on human physiological and pathological processes.

## Figures and Tables

**Figure 1 nutrients-14-02639-f001:**
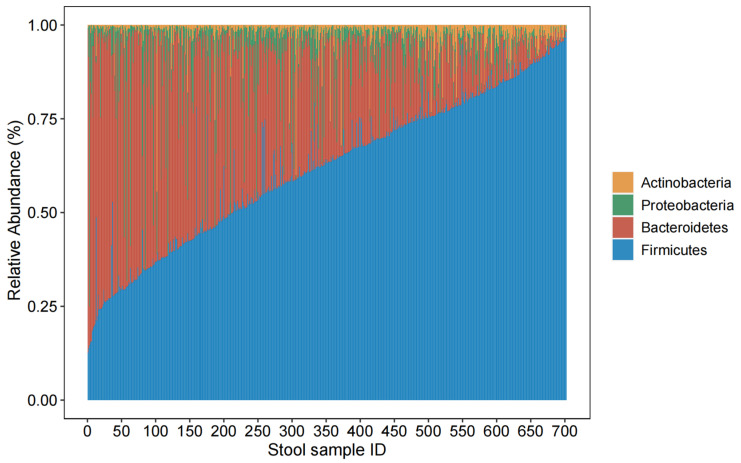
Relative abundances of the 4 most abundant phyla. Each thin vertical bar presents relative abundances determined in 1 individual stool sample, totaling 702.

**Figure 2 nutrients-14-02639-f002:**
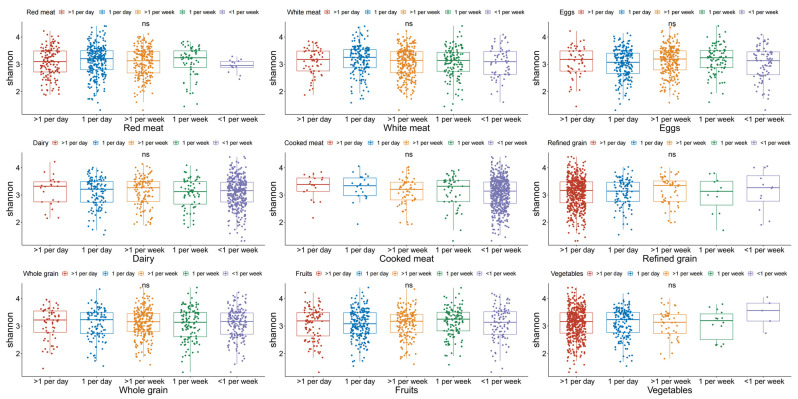
Boxplots for α-diversity Shannon index according to food intake frequencies in different food groups. ns: non-significant.

**Figure 3 nutrients-14-02639-f003:**
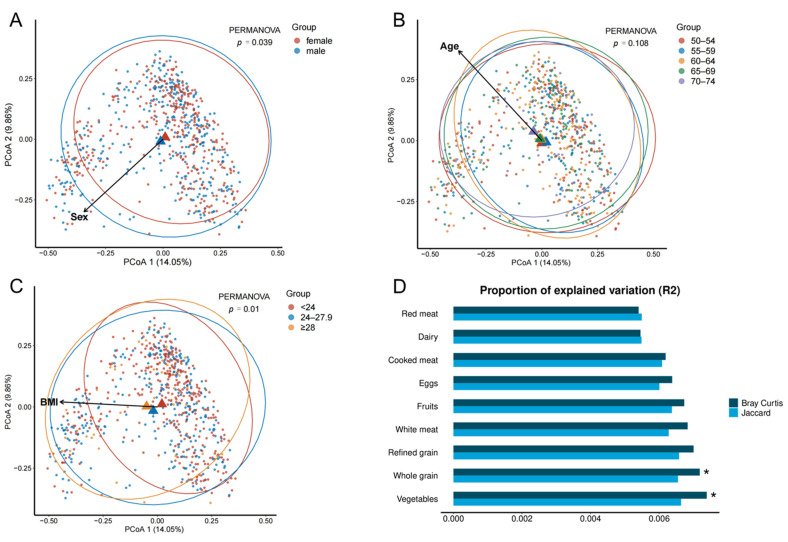
Variation in the gut microbiota composition represented by unconstrained PCoA based on the distance indexes. (**A**–**C**) present the grouping patterns of gut microbiota composition based on sex, age, and BMI. (**D**) shows percentages of variation in gut microbiota composition explained by dietary variables using multi-adjusted permutational ANOVAs (999 permutations). PCoA, principal coordinate analysis. * *p*-value < 0.05.

**Table 1 nutrients-14-02639-t001:** Characteristics of the study population (*N* = 702).

	*N*	Percentage
Sex		
Female	369	52.56%
Male	333	47.44%
Age, years		
50–54	188	26.78%
55–59	153	21.79%
60–64	181	25.78%
65–69	141	20.09%
70–74	39	5.56%
Smoking status		
Current smoker	515	73.36%
Past smoker	45	6.41%
Nonsmoker	142	20.23%
Alcohol consumption		
No	463	65.95%
Seldom	102	14.53%
Regular	137	19.52%
BMI, kg/m^2^		
<24.0	373	53.13%
24.0–27.9	282	40.17%
≥28.0	47	6.7%
Physical activity (MET, h/week)		
<33.60	175	24.93%
33.60–82.05	176	25.07%
82.05–147.80	175	24.93%
≥147.80	176	25.07%
Region		
Changsha, Hunan	190	27.07%
Hefei, Anhui	92	13.11%
Kunming, Yunnan	14	1.99%
Lanxi, Zhejiang	154	21.94%
Taizhou, Zhejiang	164	23.36%
Xuzhou, Jiangsu	88	12.54%

BMI, body mass index; MET, metabolic equivalents.

**Table 2 nutrients-14-02639-t002:** Usual dietary consumption frequencies of the study population.

Food Group	>1 per Day	1 per Day	>1 per Week	1 per Week	<1 per Week
Red meat (pork, beef, lamb, etc.)	142 (20.23%)	272 (38.75%)	211 (30.06%)	64 (9.12%)	13 (1.85%)
White meat (fish and poultry)	57 (8.12%)	170 (24.22%)	280 (39.89%)	131 (18.66%)	64 (9.12%)
Eggs	51 (7.26%)	218 (31.05%)	263 (37.46%)	88 (12.54%)	82 (11.68%)
Dairy products (milk, yoghurt, etc.)	23 (3.28%)	123 (17.52%)	121 (17.24%)	66 (9.40%)	369 (52.56%)
Cooked and cured meats (e.g., sausages)	18 (2.56%)	22 (3.13%)	57 (8.12%)	53 (7.55%)	552 (78.63%)
Refined grains (rice, flour, etc.)	521 (74.22%)	103 (14.67%)	55 (7.83%)	11 (1.57%)	12 (1.71%)
Whole grains (millet, corn, sorghum, etc.)	62 (8.83%)	107 (15.24%)	234 (33.33%)	123 (17.52%)	176 (25.07%)
Fruits	98 (13.96%)	212 (30.20%)	167 (23.79%)	127 (18.09%)	98 (13.96%)
Vegetables	480 (68.38%)	159 (22.65%)	44 (6.27%)	14 (1.99%)	5 (0.71%)

**Table 3 nutrients-14-02639-t003:** Associations between food intakes, posteriori dietary patterns, and gut microbial profiles using MaAsLins.

Food Group	Phylum	Class | Order | Family	Genus	Species	Value	Coef ^1^	Coverage (%) ^2^	Pval ^3^	Qval ^4^
Red meat	*Firmicutes*	*Bacilli | Lactobacillales | Leuconostocaceae*	*Weissella*	Uncultured organism	pd	−0.0379	28.35%	<0.0001	0.0300
Red meat	*Firmicutes*	*Bacilli | Lactobacillales | Leuconostocaceae*	*Weissella*		pd	−0.0379	29.91%	<0.0001	0.0308
Dairy	*Firmicutes*	*Clostridia | Clostridiales | Lachnospiraceae*	*Anaerostipes*	uncultured organism	pd	0.0146	67.95%	<0.0001	0.0261
Dairy	*Firmicutes*	*Clostridia | Clostridiales | Lachnospiraceae*	*Anaerostipes*		pd	0.0146	71.37%	<0.0001	0.0261
Cooked meat	*Bacteroidetes*	*Bacteroidia | Bacteroidales | Barnesiellaceae*	*Coprobacter*		pd	0.0118	11.97%	<0.0001	0.0044
Whole grains	*Firmicutes*	*Negativicutes | Veillonellales | Veillonellaceae*	*Megasphaera*	uncultured organism	mul_pd	0.0420	14.25%	<0.0001	0.0183
Refined grains	*Firmicutes*	*Bacilli | Lactobacillales | Lactobacillaceae*	*Lactobacillus*	uncultured organism	pw	0.0602	13.82%	0.0001	0.0763
Vegetables	*Firmicutes*	*Clostridia | Clostridiales | Ruminococcaceae*	*Eubacterium coprostanoligenes* group	uncultured organism	pd	−0.0767	23.50%	<0.0001	0.0123
Vegetables	*Firmicutes*	*Clostridia | Clostridiales | Ruminococcaceae*	*Eubacterium coprostanoligenes* group	uncultured organism	mul_pd	−0.0737	23.50%	<0.0001	0.0140
Vegetables	*Firmicutes*	*Clostridia | Clostridiales | Ruminococcaceae*	uncultured		pd	−0.0389	43.87%	<0.0001	0.0156
Vegetables	*Firmicutes*	*Clostridia | Clostridiales | Ruminococcaceae*	*Eubacterium coprostanoligenes* group	uncultured organism	mul_pw	−0.0746	23.50%	<0.0001	0.0173
Vegetables	*Firmicutes*	*Clostridia | Clostridiales | Ruminococcaceae*	uncultured		mul_pw	−0.0394	43.87%	<0.0001	0.0173
Vegetables	*Firmicutes*	*Clostridia | Clostridiales | Christensenellaceae*	*Christensenellaceae R7* group	uncultured organism	pd	−0.0573	27.78%	<0.0001	0.0226
Vegetables	*Firmicutes*	*Clostridia | Clostridiales | Christensenellaceae*	*Christensenellaceae R7* group	uncultured organism	mul_pd	−0.0561	27.78%	<0.0001	0.0256
Vegetables	*Firmicutes*	*Clostridia | Clostridiales | Ruminococcaceae*	*Eubacterium coprostanoligenes* group	uncultured organism	pw	−0.0754	23.50%	0.0001	0.0460
Vegetables	*Firmicutes*	*Clostridia | Clostridiales | Ruminococcaceae*	*Ruminococcaceae UCG 005*	uncultured organism	mul_pw	−0.0243	12.68%	0.0002	0.0588
Vegetables	*Firmicutes*	*Clostridia | Clostridiales | Ruminococcaceae*	uncultured		mul_pd	−0.0339	43.87%	0.0001	0.0588
Vegetables	*Firmicutes*	*Clostridia | Clostridiales | Ruminococcaceae*	uncultured		pw	−0.0388	43.87%	0.0002	0.0588
Vegetables	*Firmicutes*	*Clostridia | Clostridiales | Ruminococcaceae*	uncultured	uncultured organism	mul_pw	−0.0284	13.96%	0.0002	0.0595
Vegetables	*Firmicutes*	*Clostridia | Clostridiales | Ruminococcaceae*	*Ruminococcaceae UCG 005*	uncultured organism	pd	−0.0230	12.68%	0.0002	0.0629
Vegetables	*Firmicutes*	*Clostridia | Clostridiales | Ruminococcaceae*	uncultured	uncultured organism	pd	−0.0267	13.96%	0.0003	0.0733
Vegetables	*Firmicutes*	*Clostridia | Clostridiales | Christensenellaceae*	*Christensenellaceae R7* group	uncultured organism	mul_pw	−0.0522	27.78%	0.0003	0.0743
Vegetables	*Firmicutes*	*Clostridia | Clostridiales | Christensenellaceae*	*Christensenellaceae R7* group	uncultured organism	pw	−0.0572	27.78%	0.0003	0.0743
Vegetables	*Firmicutes*	*Clostridia | Clostridiales | Ruminococcaceae*	*Ruminococcaceae UCG 005*	uncultured organism	mul_pd	−0.0222	12.68%	0.0003	0.0743
Vegetables	*Firmicutes*	*Bacilli | Lactobacillales | Leuconostocaceae*	*Leuconostoc*	uncultured organism	mul_pw	−0.0246	17.81%	0.0003	0.0745
Vegetables	*Firmicutes*	*Bacilli | Lactobacillales | Leuconostocaceae*	*Leuconostoc*		mul_pw	−0.0246	17.81%	0.0003	0.0745
Vegetables	*Firmicutes*	*Bacilli | Lactobacillales | Leuconostocaceae*	*Leuconostoc*	uncultured organism	pd	−0.0231	17.81%	0.0005	0.0867
Vegetables	*Firmicutes*	*Bacilli | Lactobacillales | Leuconostocaceae*	*Leuconostoc*		pd	−0.0231	17.81%	0.0005	0.0867
Vegetables	*Firmicutes*	*Clostridia | Clostridiales | Ruminococcaceae*	*Ruminococcaceae UCG 005*	uncultured organism	pw	−0.0246	12.68%	0.0005	0.0900
Vegetables	*Firmicutes*	*Bacilli | Lactobacillales | Leuconostocaceae*	*Leuconostoc*	uncultured organism	mul_pd	−0.0225	17.81%	0.0006	0.0958
Vegetables	*Firmicutes*	*Bacilli | Lactobacillales | Leuconostocaceae*	*Leuconostoc*		mul_pd	−0.0225	17.81%	0.0006	0.0958
Vegetables	*Firmicutes*	*Clostridia | Clostridiales | Ruminococcaceae*	*Eubacterium* *coprostanoligenes* group		pd	−0.0804	71.23%	0.0006	0.0958
Cluster	*Firmicutes*	*Clostridia | Clostridiales | Lachnospiraceae*	*Anaerostipes*	uncultured organism	C	0.0119	67.95%	0.0001	0.0749
Cluster	*Firmicutes*	*Clostridia | Clostridiales | Lachnospiraceae*	*Anaerostipes*		C	0.0115	71.37%	0.0001	0.0858

^1^ For categorical features in MaAsLins analysis, the specific feature level for the coefficient and significance of association is reported. ^2^ Prevalence of bacterial taxa in the study sample is equal to the total of number of samples in which the feature is non-zero divided by the total number of samples used in the model. ^3^ *p*-value for MaAsLin adjusted for age, sex, BMI, smoking status, alcohol consumption, and physical activity; computed using the Maaslin2 package on R. ^4^ Corrected *p*-value by the Benjamini–Hochberg method (10% false discovery rate).

## Data Availability

Access to individual-level data, including the microbial DNA sequences encoding the 16S rRNA V4 region and associated demographic and lifestyle metadata, can be obtained upon reasonable request to the corresponding author (daimin@pumch.cn).

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
