# Peer review of "Habitual Diet Pattern Associations with Gut Microbiome Diversity and Composition: Results from a Chinese Adult Cohort"

_nutrients, 2022, doi:10.3390/nu14132639_

Round 1
Reviewer 1 Report
Authors differentiate between the transient microbiota variations in response to acute dietary intake/changes vs. the long-term effect of dietary habits on gut microbiota composition and have the objective to evaluate the latter. However, the study design which characterizes a single fecal sample does not control for the acute effect of the foods consumed in the hours preceding sample collection.
Similarly, the FFQ used - even though it is supposed to survey participants’ habitual dietary intake of the last 12 month - does not only reflect participant’s habitual intake but could also reflect rarely eaten food items (< 1 week).
This could be controlled for by a food diary (if available at sampling collection) or by comparing results with and without foods consumed < 1 week. Otherwise, this should be discussed in the limits of the paper.
Did the authors try to perform pairwise permanova (especially between BMI categories)?
Minor comments
- Figure captions appear twice
Reviewer 2 Report
This is an important study related to diet and microbiota in a large Chinese population. However I have some concerns: the subjects are considered healthy based on negative colonoscopy; however they could suffer of IBS, microscopic colitis etc or other conditions like BMI abnormalities, all able to create a bias. Therefore authors should comment this at lage in the discussionc chapter. Authors should also specify how typucal is this group of subjects for the Chinese population.
